# Bumetanide-Derived Aquaporin 1 Inhibitors, AqB013 and AqB050 Inhibit Tube Formation of Endothelial Cells through Induction of Apoptosis and Impaired Migration In Vitro

**DOI:** 10.3390/ijms20081818

**Published:** 2019-04-12

**Authors:** Yoko Tomita, Helen M. Palethorpe, Eric Smith, Maryam Nakhjavani, Amanda R. Townsend, Timothy J. Price, Andrea J. Yool, Jennifer E. Hardingham

**Affiliations:** 1Solid Tumour Group, Basil Hetzel Institute, Queen Elizabeth Hospital, Woodville South, SA 5011, Australia; yoko.tomita@sa.gov.au (Y.T.); helen.palethorpe@adelaide.edu.au (H.M.P.); eric.smith@adelaide.edu.au (E.S.); maryam.nakhjavani@adelaide.edu.au (M.N.); amanda.townsend@sa.gov.au (A.R.T.); timothy.price@sa.gov.au (T.J.P.); 2Adelaide Medical School, University of Adelaide, Adelaide, SA 5005, Australia; andrea.yool@adelaide.edu.au; 3Medical Oncology, Queen Elizabeth Hospital, Woodville South, SA 5011, Australia

**Keywords:** aquaporin 1 (AQP1), angiogenesis, AqB013, AqB050, endothelial cells, apoptosis, migration

## Abstract

AqB013 and AqB050 compounds inhibit aquaporin 1 (AQP1), a dual water and ion channel implicated in tumour angiogenesis. We tested AqB013 and AqB050 either as monotherapy or in combination on tube formation of murine endothelial cells (2H-11 and 3B-11) and human umbilical vascular endothelial cells (HUVECs). The mechanism underlying their anti-tubulogenic effect was explored by examining cell viability, induction of apoptosis and migration using 3-(4,5-dimethylthiazol-2-yl)-5-(3-carboxymethoxyphenyl)-2-(4-sulfophenyl)-2*H*-tetrazolium (MTS) assay, Annexin V/propidium iodide apoptosis assay and scratch wound assay. Tube formation of all the cell lines was inhibited by AqB013, AqB050 and the combination of the two compounds. The inhibition of 2H-11 and 3B-11 was frequently accompanied by impaired migration, whereas that of HUVEC treated with AqB050 and the combination was associated with reduced cell viability due to apoptosis. AqB013 and AqB050 exhibited an anti-tubulogenic effect through inhibition of AQP1-mediated cell migration and induction of apoptosis. Together with previously reported anti-tumour cell effect of AqB013 and AqB050, our findings support further evaluation of these compounds as potential cancer therapeutics.

## 1. Introduction

While mortality of infectious disease and vascular disease has declined in many countries, cancer has now become one of the leading causes of death and its incidence and mortality are rising worldwide. The World Health Organisation estimates that 18.1 million new cancer cases and 9.6 million cancer deaths occurred in 2018 [1]. An ongoing search for more efficacious, but less toxic, drugs against cancer has involved a paradigm shift in cancer therapeutics away from traditional cytotoxic drugs to biological therapy in the last twenty years [2]. Unlike cytotoxic drugs, which inhibit cancer cell division, biological therapies target proteins involved in the oncogenic pathway specific to the cancer phenotype.

Aquaporin 1 (AQP1), a small hydrophobic integral transmembrane protein, which spans the plasma membrane as a tetramer, was first characterised in erythrocytes and renal tubules by Denker et al. in the late 1980s and is reported to function as a water channel [3,4]. More recently, using *Xenopus laevis* oocytes injected with AQP1 cRNA, AQP1 was demonstrated to serve as a cyclic nucleotide-gated cation channel [5,6]. While AQP1 has multiple physiological roles including cerebrospinal fluid and aqueous humour secretion and regulation, maintenance of normal cytosolic osmolality, neural signal transduction, urine concentration and cell migration required for angiogenesis, it has additionally been implicated in cancer development and progression [7,8]. Over-expression of AQP1 has been observed in multiple human cancers including those of biliary tract, bladder, brain, breast, cervix, colon, nasopharynx, lung and prostate [9,10,11,12,13,14,15,16,17].

One of the proposed mechanisms underlying AQP1-mediated cancer development and growth is enhanced tumour angiogenesis. Angiogenesis is a fundamental property of cancer as it provides tumour cells with essential nutrients and oxygen required to sustain growth and an entry to systemic circulation for distant metastasis [18]. Saadoun et al. previously demonstrated delayed tumour growth and prolonged survival of AQP1 null mice, which were subcutaneously implanted with melanoma cells, and these tumours were characterised by reduced density of tumour microvessels [19]. The importance of AQP1 in tumour angiogenesis was further supported by a study in which a syngeneic melanoma murine model treated with intra-tumoural injection of AQP1 siRNA showed reduced microvessel density [20]. Another study reported AQP1 deficiency in mouse mammary tumour virus-driven polyoma virus middle T oncogene (MMTV-PyVT) mice, which spontaneously develop epithelial cancer, results in abnormal tumour microvascular anatomy and reduced vessel density [21]. These findings suggest inhibitors of AQP1 may serve as new cancer therapeutics through their activity on tumour angiogenesis in addition to their potential direct anti-tumour activity.

AqB013 and AqB050 are synthetic small molecules that were developed based on a loop diuretic bumetanide [22]. AqB013 inhibits water channel function of AQP1 [22], while AqB050 inhibits its ion channel function (A. Yool et al. manuscript in preparation). Our group previously observed impaired tube formation capacity of human umbilical vein endothelial cell (HUVEC), when treated with AqB013 [17]. The purpose of the current study was to further evaluate the anti-tubulogenic property of AqB013 and AqB050 either as monotherapy or in combination using HUVECs and two murine endothelial cell lines, 2H-11 and 3B-11. We recently demonstrated AQP1 protein expression by these three cell lines [23]. Concentrations of AqB013 and AqB050 monotherapy (40 and 80 µM) were chosen based on our previous findings where migration of HT-29 colon cancer cell line was inhibited at 80 µM, while tube formation of HUVEC line was inhibited at 40 µM by either compound [17]. We additionally tested AqB013 plus AqB050 at 20 and 40 µM each to assess if inhibiting both water and ion channel functions of AQP1 would result in enhancement of their anti-tubulogenic and anti-migratory efficacies compared to monotherapies.

## 2. Results

### 2.1. Inhibition of Tube Formation by AqB013 and AqB050

The effect of AqB013 and AqB050 either as monotherapy or in combination was assessed in tube formation assays using 2H-11, 3B-11 and HUVEC lines (Figure 1). For 2H-11 line, AqB050 was more efficacious than AqB013. For AqB013 and AqB050, the number of loops formed was reduced by 19% (*p* = 0.0182) and 58% (*p* < 0.0001) at 40 µM, respectively, and 57% (*p* < 0.0001) and 86% (*p* < 0.0001) at 80 µM, respectively, compared to the vehicle control. Combining the two compounds achieved equivalent inhibition to AqB050 alone; tube formation was reduced by 61% for AqB013 plus AqB050 at 40 µM each (*p* = 0.0001), designated 40/40 µM. For 3B-11, tube formation was reduced similarly by AqB013 and AqB050: 65% (*p* < 0.0001) and 78% (*p* < 0.0001) inhibition was observed, respectively, at 80 µM compared to the vehicle control. Combination treatment at 40/40 µM resulted in reduction by 84% (*p* < 0.0001). The HUVEC line was more sensitive to AqB013 than AqB050 and no tube formation was observed for AqB013 at 80 µM (*p* < 0.0001), while AqB050 at 80 µM caused 64% inhibition (*p* < 0.0001). Fifty-five per cent inhibition was achieved by the combination treatment at 40/40 µM. 

### 2.2. Effect of AqB013 and AqB050 on Cell Viability

To examine the mechanism underlying reduced tube formation of endothelial cells when treated with AqB013 and AqB050 either as monotherapy or in combination, cell viability was assessed using MTS assay (Figure 2). For 2H-11 and 3B-11 lines, minimal or no inhibition of cell viability was observed. Treatment with AqB050 at 80 µM reduced cell viability of 2H-11 line by 16% (*p* = 0.0006). 3B-11 line demonstrated no inhibition of cell viability with any of the treatments tested. In contrast, cell viability of HUVEC line was reduced by 21% for AqB050 at 40 µM (*p* = 0.0004), 48% for AqB050 at 80 µM (*p* < 0.0001), 17% for AqB013 plus AqB050 at 20/20 µM (*p* = 0.0041) and 31% for AqB013 plus AqB050 at 40/40 µM (*p* < 0.0001).

### 2.3. Induction of Apoptosis by AqB050

To further elucidate the cause of impaired cell viability induced by the compounds, induction of apoptosis was assessed in the HUVEC line treated with AqB050 either as monotherapy or in combination with AqB013. A reduction in cell viability was demonstrated. An increase in apoptosis was observed with treatment containing AqB050 at 40 µM or higher doses. Compared to the vehicle control, the proportion of cells in apoptosis increased by 25% with 40 µM AqB050, 32% with 80 µM AqB050, and 29% with AqB013 plus AqB050 at 40/40 µM (Figure 3).

### 2.4. Inhibition of Migration by AqB013 and AqB050

Anti-migratory actions of AqB013 and AqB050 either as monotherapy or in combination were assessed on scratch wound assay using 2H-11, 3B-11 and HUVEC lines (Figure 4). The lowest doses of compounds required to inhibit wound closure of 2H-11 line compared to the vehicle control at 16 h were AqB013 at 80 µM (37%; *p* = 0.0002), AqB050 at 40 µM (56%; *p* < 0.0001) and AqB013 plus AqB050 at 20/20 µM (52%; *p* < 0.0001). Wound closure in the 3B-11 line was similarly inhibited by the compounds; however, the magnitude of inhibition was smaller than that for the 2H-11 line. Closure rate was reduced by 23% for AqB013 at 80 µM (*p* = 0.0035), 23% and 26% for AqB050 at 40 µM (*p* = 0.0037) and 80 µM (*p* = 0.001), respectively, and 33% for AqB013 plus AqB050 at 40/40 µM (*p* < 0.0001), compared to the vehicle control. The most prominent inhibition of wound closure was observed in the HUVEC line. Unlike the murine endothelial cell lines, AqB013 did not reduce wound closure of HUVEC line even at 80 µM and no statistically significant difference was observed at AqB013 at 40 and 80 µM compared to the vehicle control. AqB050 at 40 and 80 µM reduced wound closure by 89% (*p* < 0.0001) and 94% (*p* < 0.0001), respectively, and AqB013 plus AqB050 at 20/20 and 40/40 µM reduced wound closure by 34% (*p* = 0.0045) and 91% (*p* < 0.0001), respectively, compared to the vehicle control. 

## 3. Discussion

This is the first report of the effect of AqB013 and AqB050 combined, showing anti-tubulogenic properties, extending our previous report of the effect of AqB013 in inhibiting HUVEC tube formation [17]. In the current study, using HUVEC and two murine endothelial cell lines, we confirmed our previous finding that AqB013 inhibits tube formation of endothelial cells in a dose dependent manner without impairing their cell viability. At 80 µM, AqB013 reduced tube formation by 57% and 65% for 2H-11 and 3B-11 lines, respectively, and completely inhibited that of HUVEC line. AqB050 and AqB013 plus AqB050 were similarly shown to reduce tube formation of the three cell lines again in a dose dependent manner. Combining AqB013 and AqB050 appeared to potentiate their anti-tubulogenic effect for 3B-11 and HUVEC lines. For 3B-11, only AqB050 at 40 µM, but not AqB013 at 40 µM, resulted in inhibition of tube formation by 32%, whereas AqB013 plus AqB050 at 40/40 µM, resulted in 84% inhibition of tube formation compared to the vehicle control. Similarly, for HUVEC line, AqB013 plus AqB050 40/40 µM resulted in 55% inhibition of tube formation, while as monotherapy neither compound at 40 µM demonstrated anti-tubulogenic effect compared to the vehicle control.

The anti-tubulogenic effect of AqB050 was accompanied by impaired cell viability. The reduced tube formation induced by AqB050 at 80 µM in the 2H-11 and HUVEC lines was accompanied by a significant decrease in cell viability with HUVEC line being particularly susceptible to the effect. No impaired cell viability was observed for 3B-11, while AqB050 at 80 µM inhibited its tube formation. This difference in the sensitivity to cytotoxicity of AqB050 between the HUVEC primary cell line and 2H-11 and 3B-11 cell lines may relate to the simian virus 40-transformed nature of 2H-11 and 3B-11 cell lines, to make them immortal. SV40 transformation is mediated largely by the interaction of its T-antigen with tumour suppressor proteins p53 and pRb of transformed cells [24]. Blocking these pathways results in a switch to cell cycling thus making the cells more resistant to apoptosis [25].

The increase in annexin-V staining following treatment of HUVECs with AqB050 at doses which inhibited tube formation suggests apoptosis was at least partly involved in the anti-tubulogenic action of AqB050 on this cell line. A link between AQP1 and apoptosis has been suggested previously. Wu et al. reported shRNA-mediated knockdown of AQP1 in two osteosarcoma lines U2OS and MG63 resulted in promotion of apoptosis with reduction in the level of antiapoptotic protein BCL-2 and increase in the levels of proapoptotic protein Bax and cleaved caspase 3 [26]. Additionally, NIH-3T3 murine fibroblast cell line transfected with AQP1 expression constructs demonstrated decreased apoptosis compared to the mock transfection [12]. In vivo, reduced angiogenesis mediated by AQP1 inhibition may additionally contribute to induction of tumour cell apoptosis through hypoxia. Mouse melanoma xenograft model treated with AQP1 siRNA demonstrated decreased microvessel density and increased CASP3 and HIF-1α expression [20,27]. An increase in AQP1 expression of resected gastric adenocarcinoma correlated with an increase in apoptosis [28]. Given we observed no impairment of viability in HUVEC treated with AqB013, the pro-apoptotic effect of AQP1 inhibition with AqB050 may be specifically mediated by the blockade of the ion channel, but not water channel function of AQP1. Inhibition of aquaporins using mercury, which is known to block water permeation through AQP1, has been suggested to block apoptotic volume decrease (AVD), one of the earliest and highly conserved events in apoptosis [29]. Regulation of cell volume and its ionic homeostasis is critical in the activation and repression of apoptosis and AVD has been linked to alteration in intracellular ions, in particular, loss of intracellular potassium ions [30]. AQP1 shares similar structural organisation to potassium channels and functions as a non-selective cation channel permeant to potassium, sodium and cesium, in addition to being an osmotic water channel [5,31]. Evasion of apoptosis is a known hallmark of cancer with pharmacological manipulation of potassium channels being reported to induce apoptosis of various tumour cells in vitro [30]. Treatment with AqB050 may cause disruption of intracellular potassium homeostasis in tumour cells inducing apoptosis and resultant cell death and this requires further exploration. AqB050 may also exert its effect through unreported non-AQP1 targets.

Treatment of HUVECs with the lower dose combination of AqB013 plus AqB050 at 20/20 µM, where tube formation was inhibited by 28%, was not associated with an increase in the annexin-V staining. As this combination treatment resulted in 17% inhibition of cell viability, the impaired tube formation observed can be explained by its anti-proliferative effect through mechanisms other than induction of apoptosis. Additionally, AqB013 plus AqB050 at 20/20 µM reduced wound closure of HUVECs by 34% compared to the vehicle control, suggesting inhibition of two-dimensional migration by this combination treatment partly contributed to the reduced tube formation. In fact, for the murine endothelial cell lines, inhibition of tube formation seen with AqB013 and AqB050 either as monotherapy or in combination was frequently accompanied by impaired wound closure in the absence of reduced cell viability. Treatment of 2H-11 line with AqB013 at 80 µM, AqB050 at 40 µM and 80 µM, and AqB013 plus AqB050 at 40/40 µM, which inhibited tube formation by more than 50%, all reduced its two-dimensional migration. Similar observations were made with 3B-11; treatment with AqB013 at 80 µM, AqB050 at 80 µM and AqB013 plus AqB050 at 40/40 µM was all associated with greater than 50% inhibition of tube formation and reduced two-dimensional migration. It is plausible that AqB013 and AqB050 exhibit their anti-tubulogenic effect through inhibition of cell motility for these cell lines. This is supported by the finding that AQP1-deficient endothelial cells generated from the aorta of AQP1-null mice demonstrated impaired cell migration and invasion, compared to those from AQP1-wild type mice [19]. Furthermore, siRNA-mediated silencing of AQP1 in human microvascular endothelial cell-1 (HMEC-1) was shown to result in a lack of F-actin polarisation at the leading edge of the plasma membrane and failure of these cells to organise a cord-like network in vitro [32]. 

In contrast to the murine cell lines, HUVEC line treated with AqB013 at 80 µM showed complete inhibition of tube formation in the absence of impaired two-dimensional migration or cell viability, supporting an alternative underlying mechanism. As extracellular matrix was present in the tube formation assay, but not in the scratch wound assay, we postulate this unusual discordance between the effect of AqB013 on cell motility and tube formation derives from its effect on extracellular matrix (ECM) degradation. Using a syngeneic melanoma mouse model, Simone et al. found intra-tumoural injection of AQP1 siRNA reduced expression of matrix metallopeptidase-2 (MMP-2), one of the key regulators of tumour angiogenesis [27]. Treatment of NSCLC cell lines LLC and LTEP-A2 with AQP1 siRNA and the resultant knockdown of the expression have also been reported to reduce transwell migration and invasion as well as wound closure and was accompanied by reduced expression of MMP-2 and metallopeptidase-9 (MMP-9) [33]. Impaired ECM degradation secondary to AqB013 could possibly explain the inhibited tube formation of HUVEC line observed with with AqB013 at 80 µM, while two-dimensional migration remained unaffected by the treatment.

In the current study, AqB050, the ion channel inhibitor, appeared more effective in impairing two-dimensional migration of endothelial cells than the water channel inhibitor AqB013. AqB050 at 40 and 80 µM inhibited wound closure of all the endothelial cell lines tested, whereas AqB013-mediated impaired wound closure was only observed for murine cell lines at 80 µM with the magnitude of inhibition by AqB050 at 80 µM being larger than that of AqB013 at 80 µM. Similarly, we observed treatment with AqB013 plus AqB050 at 40/40 µM resulted in the equivalent inhibition of wound closure to AqB050 at 40 µM for all the endothelial cell lines. Both water and ion channel functions of AQP1 have been suggested to play a part in cell motility. Jiang demonstrated reduced water permeability induced by AQP1 RNA-interference was associated with decelerated wound closure of HT-20 human colon cancer cell line [34]. AqB011, another bumetanide-derived potent AQP1 ion channel inhibitor with no effect on its water channel activity at up to 200 µM, was shown to impair wound closure of AQP1 expressing HT-29 human colon cancer cell line, however, the finding was not replicated in SW-480, another human colon cancer cell line which expresses AQP5, but not AQP1 [35].

There have been two publications describing the anti-cancer efficacy of AqB013 and AqB050 in vitro. Our group previously reported that wound closure and invasion of moderately AQP1-expressing colon cancer cell line HT-29 was inhibited by AqB013 [17]. Similarly, Klebe et al. showed treatment of malignant mesothelioma (MM) cell line H226 and primary MM cells harvested from patients’ pleural effusion with AqB050 impaired their cell proliferation, migration and anchorage-independent cell growth [36]. Together with the current study, which suggests an anti-tubulogenic effect of AqB013 and AqB050, these findings support the potential therapeutic efficacy of these bumetanide-derived AQP1 inhibitors. 

## 4. Materials and Methods

### 4.1. Cell Lines and Culture

Human umbilical vein endothelial cell (HUVEC) and two murine endothelial cell lines (2H-11 and 3B-11) were purchased from American Type Culture Collection (ATCC, Manassas, VA, USA). 2H-11 and 3B-11 lines were maintained in Dulbecco’s Modified Eagle Medium (DMEM; Life Technologies, Carlsbad, CA, USA), supplemented with 10% foetal bovine serum (FBS; Corning, Corning, NY, USA), 2 mM L-alanyl-L-glutamine dipeptide (GlutaMAX^TM^; Life Technologies) and 50 U/mL-50 µg/mL penicillin–streptomycin (Life Technologies). HUVEC line was maintained in Vascular Cell Basal Medium supplemented with Endothelial Cell Growth Kit-VEGF (ATCC). Cultures were maintained in 5% CO_2_ at 37 °C.

### 4.2. Preparation of Inhibitors 

Stock solutions of custom synthesised compounds AqB013 and AqB050 (Spacefill Enterprises LLC, Oro Valley, AZ, USA) were prepared by resuspending each compound in dimethyl sulphoxide (DMSO; Sigma-Aldrich, St Louis, MO, USA) at 40 mM and this was diluted in the respective complete medium to achieve the indicated concentrations for the experiment. Vehicle control was prepared by adding DMSO to the complete medium at the dilution equivalent to that for the highest dose of compounds assessed. 

### 4.3. Apoptosis Assay

HUVECs were seeded at 1 × 10^5^ cells per well on 6-well plates and incubated for 48 h in 5% CO_2_ at 37 °C. Apoptosis controls were treated with Bacopaside II (Sigma-Aldrich) at 20 µM for 24 h prior to harvest. Necrosis controls were prepared by heating harvested cells to 63 °C for 30 min. Cells were treated with endothelial growth medium, vehicle (DMSO), AqB050 at 40 and 80 µM or AqB013 plus AqB050 at 20/20 and 40/40 µM for 24 h. Harvested cells were stained with Annexin-V-FLUOS staining kit (Roche Diagnostics, Mannheim, Germany) as per the manufacturer’s instruction and fluorescence was read on BD FACSCanto II cell analyser (BD Biosciences, San Jose, CA, USA), and analysed using FlowJo software (FlowJo, Ashland, OR, USA) v10.4 as described previously [37].

### 4.4. MTS Viability Assay

Cells were seeded at 1 × 10^4^ cells per well on 96-well plate in a respective complete medium and incubated overnight in 5% CO_2_ at 37 °C. The medium was replaced by fresh complete medium containing various concentrations of the compounds with DMSO as the vehicle control. Following further incubation for 24 h, viability was assessed using CellTiter 96® AQueous Non-Radioactive Cell Proliferation Assay (Promega, Madison, WI, USA) according to the manufacturer’s instruction. The absorbance was read at 492 nm and the results were expressed as absorbance normalised to the vehicle control.

### 4.5. Scratch Wound Assay

Cells were plated on 96-well plate in the respective complete medium at 1.2 × 10^4^, 4 × 10^4^ and 3.5 × 10^4^ cells per well for 2H-11, 3B-11 and HUVEC lines, respectively. After 6 h of incubation in 5% CO_2_ at 37 °C, the medium was replaced with DMEM-based media supplemented with 2% FBS for 2H-11 and 3B-11 lines and HUVEC growth medium for HUVEC line, both containing 1 µg/mL mitomycin C (Sigma-Aldrich). These media are referred to as “mitomedia”. A scratch wound was made on the cell monolayer and cells were incubated for 1 h in the fresh mitomedia. The medium was then changed to mitomedia containing various concentrations of compounds with DMSO as a vehicle. Area surface of the wound was serially monitored and images were obtained for the next 16 h on Eclipse TE2000-U light microscope (Nikon, Tokyo, Japan) at 40× magnification. The wound area was quantified using NIS-Elements BR software (Nikon) and the result was expressed as relative wound closure (%) compared to time point 0.

### 4.6. Tube Formation Assay

Cells were plated on a 96-well angiogenesis µ-plate (Ibidi, Martinsried, Germany) coated with Matrigel (Corning) according to the supplied protocol in the respective medium containing various concentration of compounds with DMSO as a vehicle. The number of loops formed was counted at 2 h.

### 4.7. Statistical Analysis

Graph Pad Prism was used for statistical analysis. A one-way ANOVA with Tukey’s multiple comparisons test was carried out. Statistical significance was accepted at *p* < 0.05.

## 5. Conclusions

The current study demonstrated two bumetanide-derived inhibitors of AQP1, AqB013 and AqB050, inhibit tube formation in vitro using murine and human endothelial cell lines known to express AQP1, supporting their potential utility as cancer therapeutics. Their anti-angiogenic efficacy was suggested to be at least partly driven by inhibition of cell migration and induction of apoptosis where AQP1 has been reported to play a role. Additionally, we postulate inhibition of AQP1 water channel by AqB013 resulted in impaired AQP1-mediated degradation of ECM, possibly explaining the inhibited tube formation of HUVEC line observed with this compound, where two-dimensional migration measured using scratch wound assay was preserved. Further research is required to continue characterising the compounds with respect to their mechanisms of actions and their in vivo efficacy.

## Figures and Tables

**Figure 1 ijms-20-01818-f001:**
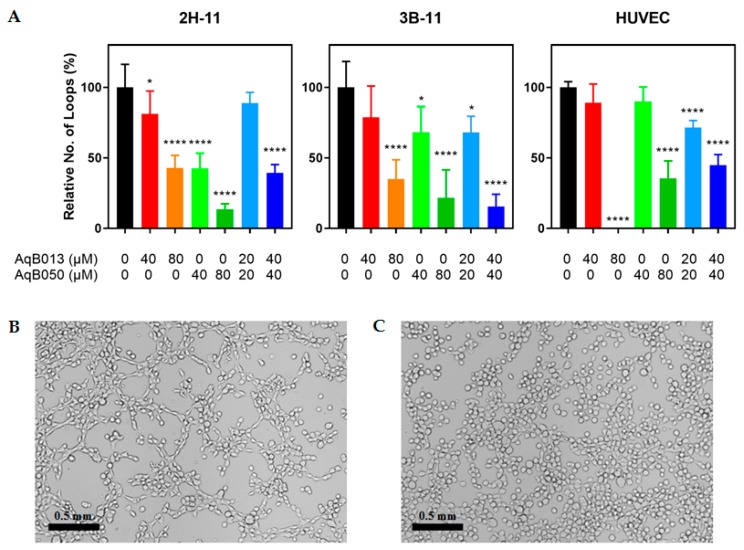
Inhibition of tube formation by AqB013 and AqB050. 2H-11, 3B-11 and HUVEC lines were treated with various amount of AqB013 and/or AqB050 for 2 h and the number of loops formed was counted. (**A**) Results of six replicates are shown as the mean loop formation normalised to the vehicle control. The error bar represents SD and significant difference as compared to the vehicle control is indicated by asterisks (* *p* < 0.05 and **** *p* < 0.0001). (**B**,**C**) Representative images of 2H-11 line treated with (**B**) the vehicle control and (**C**) AqB050 at 80 µM. Magnification = 100× (scale bar = 0.5 mm).

**Figure 2 ijms-20-01818-f002:**
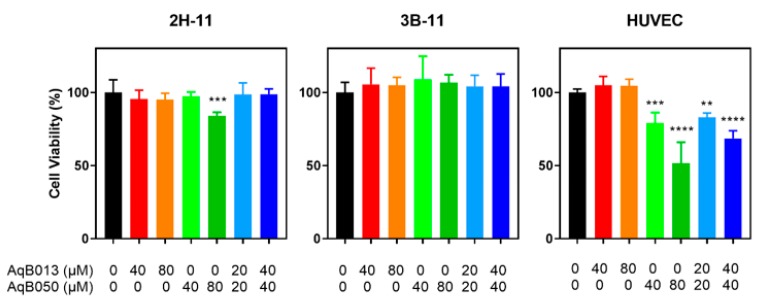
Inhibition of cell viability by AqB013 and AqB050. Cells were treated with AqB013 and AqB050 for 24 h, and viability was determined by MTS assay. The mean of five replicates normalised to the vehicle control is shown. The error bar represents SD and the significant difference to the vehicle control is indicated by asterisks (** *p* < 0.01, *** *p* < 0.001 and **** *p* < 0.0001).

**Figure 3 ijms-20-01818-f003:**
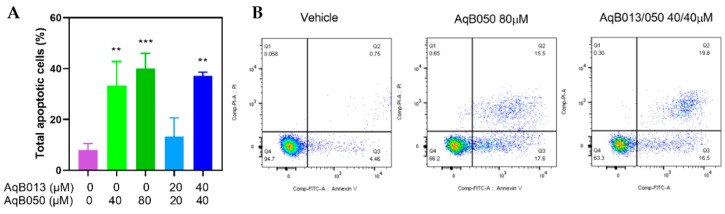
Induction of HUVEC line apoptosis by AqB050 and AqB013 plus AqB050. HUVEC cells treated with AqB050 monotherapy and AqB013 plus AqB050 for 24 h were analysed for apoptosis and necrosis using Annexin V/propidium iodide apoptosis assay. (**A**) In the bar graph, results of three replicates are shown as mean percentages of apoptotic cells with the error bar representing SD. The significant difference to the vehicle control is indicated by asterisks (** *p* < 0.01 and *** *p* < 0.001). (**B**) The scatter plots show population gates of viable cells (left lower quadrant), and cells in early apoptosis (right lower quadrant), late apoptosis (right upper quadrant), and necrosis (left upper quadrant).

**Figure 4 ijms-20-01818-f004:**
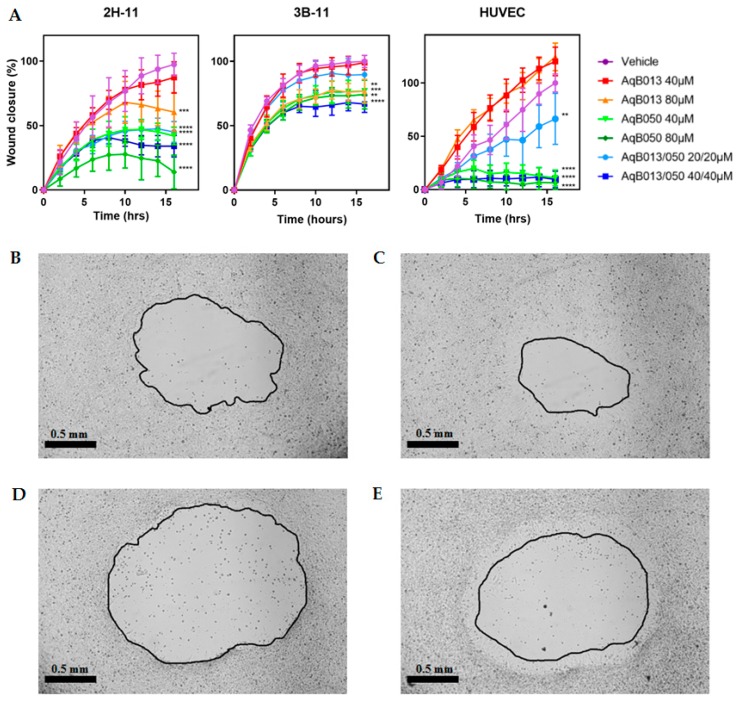
Inhibition of scratch wound closure by AqB013 and AqB050. Scratch wound for 2H-11, 3B-11 and HUVEC lines treated with AqB013 and/or AqB050 were serially imaged and wound closure was expressed relative to that at Time 0. (**A**) Results of five replicates are shown as the mean percentage wound closure normalised to the vehicle control. The error bar represents SD and the significant difference to the vehicle control is indicated by asterisks (** *p* < 0.01, *** *p* < 0.001 and **** *p* < 0.0001). (**B**–**E**) Representative images of 2H-11 treated with the vehicle control at (**B**) 0 h and (**C**) 10 h, and AqB050 at 80 µM at (**D**) 0 h and (**E**) 10 h at 40× magnification (scale bar = 0.5 mm).

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
