# Peer review of "Bumetanide-Derived Aquaporin 1 Inhibitors, AqB013 and AqB050 Inhibit Tube Formation of Endothelial Cells through Induction of Apoptosis and Impaired Migration In Vitro"

_ijms, 2019, doi:10.3390/ijms20081818_

Round 1

Reviewer 1 Report

The authors in this article describe a potential role of two AQP1 inhibitors in endothelial cell function. The results are interesting but extra controls and one key experiment would improved the quality of the article.

Major points :

- an experiment of real endothelial tube sprouting has to be performed, as in fibrin gel (please check this method : http://4designbiosciences.com/pubs/Nakatsu%20and%20Hughes%20(MIE).pdf).

- Can the authors explain why no apoptosis is observed in Figure 3A for the condition 20/20 when viability is decreased in Figure 2 ?

Minor points :

- images of "tube" formation have to be shown in Figure 1 and images of endothelial cell migartion in Figure 4.

Author Response

Response to Reviewer 1 comments;

Point 1: an experiment of real endothelial tube sprouting has to be performed, as in fibrin gel (please check this method : http://4designbiosciences.com/pubs/Nakatsu%20and%20Hughes%20(MIE).pdf). 

Response 1: We appreciate a suggestion to test our two AQP1 inhibitors on endothelial cells using an in vitro fibrin bead assay which replicates the major steps of angiogenesis more closely than tube formation assay that we performed. In the updated manuscript we were unable to provide any additional results based on the suggested fibrin bead assay as we have no experience in performing the assay and it would require a major extension in the deadline set for revising the manuscript. Matrigel tube formation assay is a widely utilised in vitro assay to rapidly assess pro and anti-angiogenic compounds and although it does not model the entire complex interactions found in in vivo, it still considers the key steps of angiogenesis, namely endothelial cell migration, protease activity and tubule formation.

Point 2: can the authors explain why no apoptosis is observed in Figure 3A for the condition 20/20 when viability is decreased in Figure 2 ? 

Response 2: Discussion on  potential reasons behind lack of increased apoptosis seen with HUVECs at AqB013 plus AqB050 at 20/20µM is added (Line 202-).

Point 3: images of "tube" formation have to be shown in Figure 1 and images of endothelial cell migartion in Figure 4.

Response 3: Representing images from 2H-11 tube formation and scratch wound assays were included in the results (Figure 1 and 4).

Reviewer 2 Report

In the manuscript, Tomita and colleagues showed that two inhibitors of aquaporin 1, AqB013 andAqB050, alone or in combination, inhibited the capability of endothelial cells to perform tubes like capillaries when seeded on Matrigel. Moreover, in order to identify the mechanisms, they showed that both compounds significantly affected endothelial cell survival (in HUVEC) and migration (in all cells).

The manuscript is concise and well written. The data are convincing and 

support the conclusions.

Concerns

1. Please justify the concentrations used in the assays

2. Explain the rationale of the combination and the results obtained (did the authors observe some potentiation? synergism?)

3. Can the authors present the most significant picture of

matrigels?  (fig. 1).

4. Fig. 1. In materials and methods, the authors said to measure the loops after two hours from endothelial cells seeding on Matrigel. Is that correct? Also for HUVEC? These cells need more than 2 hours to perform sprouting

5. Tube formation on matrigel is not a reliable method to measure angiogenesis (Angiogenesis (2018) 21:425–532; https://doi.org/10.1007/s10456-018-9613-x). Please rephrase the statements on the antiangiogenic activity of the two inhibitors throughout the manuscript

6. Figure 4. the pharmacological effect of combination seems to be dependent on B050 40uM in each cell lines. This aspect should be discussed

7. Discussion line 168: NIH3T3 are not NSCLC. Please correct this.

8. Discussion line 186: inhibition of ion channel of aquaporin 1 by the two inhibitors seems to play a role not only in survival but also migration (see fig 4). Discuss this aspect

9. Material and methods: line261: scratch assay: did you performed the scratch on 96 well plates? It is correct? Can you please show the most representative effect of inhibitors on scratch by pictures in fig 4?

Author Response

Response to Reviewer 2 comments;

Point 1: Please justify the concentrations used in the assays

Response 1:  Justification for the concentrations used is now discussed at the end of the introduction (Line 72-).

Point 2: Explain the rationale of the combination and the results obtained (did the authors observe some potentiation? synergism?).

Response 2: The rational for combining the two AQP1 inhibitors was included at the end of the introduction and their potential synergism was mentioned in the discussion (Line 75-and Line 159-).

Point 3: Can the authors present the most significant picture of

matrigels?  (fig. 1).

Response 3: Representing images from 2H-11 tube formation assay are included in Figure 1.

Point 4: Fig. 1. In materials and methods, the authors said to measure the loops after two hours from endothelial cells seeding on Matrigel. Is that correct? Also for HUVEC? These cells need more than 2 hours to perform sprouting

Response 4: Our HUVEC line was purchased directly from ATCC and tube formation assay was performed using these cells within 4 passages. During the optimisation run, untreated HUVECs demonstrated the most extensive tube formation after 2 hours of incubation.

Point 5: tube formation on matrigel is not a reliable method to measure angiogenesis (Angiogenesis (2018) 21:425–532; https://doi.org/10.1007/s10456-018-9613-x). Please rephrase the statements on the antiangiogenic activity of the two inhibitors throughout the manuscript.

Response 5: Statements referring to anti-angiogenic activity of AqB013 and AqB050 are rephrased to refer to their anti-tubulogenic activity.

Point 6: Figure 4. the pharmacological effect of combination seems to be dependent on B050 40uM in each cell lines. This aspect should be discussed.

Response 6: Discussion on results of scratch wound assay with combined AqB013 and AqB050, compared to AqB050 alone is added (Line 237-).

Point 7: Discussion line 168: NIH3T3 are not NSCLC. Please correct this.

Response 7: Description of NIH3T3 cell line was corrected (Line 181).

Point 8: Discussion line 186: inhibition of ion channel of aquaporin 1 by the two inhibitors seems to play a role not only in survival but also migration (see fig 4). Discuss this aspect.

Response 8:  Discussion on the role of AQP1 ion channel inhibition in impairing cell motility is added (Line 243-).

Point 9: Material and methods: line261: scratch assay: did you performed the scratch on 96 well plates? It is correct? Can you please show the most representative effect of inhibitors on scratch by pictures in fig 4?

Response 9: Scratch wound assay was performed on 96-well plates and representing images for 2H-11 are included in Figure 4.

Round 2

Reviewer 1 Report

I appreciate the additional informations in the new version of this article.

I understand the reasons for not developing the experiment from point 1.

This article reached good quality, which I believe, is adapted to IJMS.